# Optimistic Algorithms for
# Adaptive Estimation of the Average Treatment Effect

**Ojash Neopane** [1]   **Aaditya Ramdas** [1 2]   **Aarti Singh** [1]

## Abstract

Estimation and inference for the Average Treatment Effect (ATE) is a cornerstone of causal inference and often serves as the foundation for developing procedures for more complicated settings. Although traditionally analyzed in a batch setting, recent advances in martingale theory have paved the way for adaptive methods that can enhance the power of downstream inference. Despite these advances, progress in understanding and developing adaptive algorithms remains in its early stages. Existing work either focus on asymptotic analyses that overlook exploration-exploitation tradeoffs relevant in finite-sample regimes or rely on simpler but suboptimal estimators. In this work, we address these limitations by studying adaptive sampling procedures that take advantage of the asymptotically optimal Augmented Inverse Probability Weighting (AIPW) estimator. Our analysis uncovers challenges obscured by asymptotic approaches and introduces a novel algorithmic design principle reminiscent of optimism in multi-armed bandits. This principled approach enables our algorithm to achieve significant theoretical and empirical gains compared to previous methods. Our findings mark a step forward in the advancement of adaptive causal inference methods in theory and practice.

## 1. Introduction

The problem of estimating the average treatment effect (ATE) is central to causal inference and has been extensively studied. We have a precise understanding of the difficulty of this problem in both asymptotic and nonasymptotic regimes.

However, our understanding of the challenges associated with *adaptive* ATE estimation remains limited.

Classically, adaptive ATE estimation has been analyzed in an asymptotic setting, where past work has focused on designing adaptive sampling procedures that ensure that the resulting ATE estimate achieves the smallest possible asymptotic variance, that is, the semiparametric efficiency bound. More recently, there has been growing interest in developing algorithms that provide nonasymptotic performance guarantees. However, these works suffer from certain drawbacks that lead to poor finite sample performance, an issue that we discuss in detail in Sections 2 and 4.1.

In this work, we take a nonasymptotic perspective on adaptive ATE estimation, focusing on the Augmented Inverse Propensity Weighting (AIPW) estimator. Our finite-sample analysis reveals key aspects of algorithmic design that prior work has overlooked. This enables us to propose a new algorithm with substantially improved theoretical and empirical performance while also simplifying the analysis.

At the heart of our approach is the insight that initially over-sampling arms that should eventually be under-sampled according to the (unknown) optimal allocation can lead to better estimates of the ATE. Interestingly, this idea can be interpreted as an instance of the principle of *optimism*, a well-established algorithmic design paradigm in the literature on regret minimization in multi-armed bandits (MAB) and reinforcement learning. We discuss this connection in more detail in Section 4.

**Contributions.**   Our main contributions are as follows:

1. We develop and analyze a new algorithm, Optimistic Policy Tracking (`OPTrack`), for the adaptive estimation of ATE that enjoys significant theoretical improvements over previous approaches along with a significantly simplified analysis.

2. We perform simulations that demonstrate that our theoretical improvements translate into empirical improvements, especially in the small sample regime, which is critical for applications such as randomized clinical trials.

[1]Machine Learning Department, Carnegie Mellon University [2]Department of Statistics & Data Science, Carnegie Mellon University. Correspondence to: Ojash Neopane <oneopane@andrew.cmu.edu>.

*Proceedings of the 42$^{nd}$ International Conference on Machine Learning*, Vancouver, Canada. PMLR 267, 2025. Copyright 2025 by the author(s).

**Organization.** The remainder of this paper is structured as follows. In Section 2, we review previous and related work. Section 3 introduces our problem setup, establishing the necessary framework for our contributions. In Section 4, we identify key limitations of existing approaches and present our proposed `OPTrack` algorithm. Section 5 contains our main theoretical results, providing a rigorous performance characterization of `OPTrack`. Finally, in Section 6, we empirically validate our method, demonstrating its superior performance compared to existing approaches and its competitiveness with—often surpassing—even an infeasible oracle baseline.

## 2. Prior and Related Works

Adaptive experimental design has a long and distinguished history, dating back to the seminal work of Neyman (1934) on optimal allocation in experimental studies. Thompson (1933) introduced a Bayesian adaptive design, thus laying the foundation for the MAB problem. Thompson's approach of sequential updating beliefs about treatments (or arms) based on observed outcomes is now central in MAB research (Lattimore & Szepesvári, 2020). However, many problem formulations focus on maximizing cumulative rewards over repeated rounds of exploration-exploitation. In contrast, our objective of ATE estimation differs from the typical MAB focus and raises different forms of exploration trade-offs.

### 2.1. Prior Work

Our work builds on a recent line of work investigating adaptive algorithms aimed at efficiently estimating ATE. Hahn et al. (2009) sparked this research direction by proposing a two-stage design, conceptually similar to the Explore-then-Commit algorithms in MAB (Garivier et al., 2016) and showing that it asymptotically attains the minimum-variance semiparametric efficiency bound. Subsequently, Kato et al. (2020) introduced a fully adaptive procedure using the *adaptive* AIPW estimator (`A2IPW`), and showed that it is asymptotically optimal (in the above sense) while also providing improved empirical performance compared to the less adaptive two-stage design. Later, Cook et al. (2024) proposed an alternative method called Clipped Standard-Deviation Tracking (`ClipSDT`), which inherits the same asymptotic optimality under milder assumptions, admits modern uncertainty quantification tools (Waudby-Smith et al., 2022), and outperforms the earlier approach empirically. In parallel work, Li et al. (2024) significantly generalized the two-stage design in Hahn et al. (2009), extending its applicability to a broad spectrum of problems, including Markovian and non-Markovian decision processes.

Despite these advances, all of the above approaches focus on characterizing the asymptotic behavior of their approaches, leaving open questions about finite-sample performance of their work. In order to address these questions, Dai et al. (2023) takes an initial step toward understanding the nonasymptotic difficulty by introducing the `ClipOGD` algorithm for the fixed-design setting. They introduce and analyze the Neyman regret (in the design-based setting), which is a normalized proxy to the variance of the resulting ATE estimate. Even more recently, Neopane et al. (2025) proposed and analyzed the `ClipSMT` algorithm for the superpopulation setting and shows that it enjoys an improved $\log T$ bound on the Neyman regret.

Although these two works take important first steps toward understanding the nonasymptotic difficulty of adaptive ATE estimation, their algorithms rely on the `IPW` estimator which is known to be suboptimal. In fact, these works define the Neyman regret with respect to the minimum variance `IPW` estimator, where the minimization is performed over all possible allocations. In contrast, our definition of the Neyman regret is much stronger as the baseline against which we compete is defined as the minimum attainable variance over *all* pairs of estimators and allocations. Notably, using this stronger definition of regret, the aforementioned approaches obtain linear Neyman regret, where as we are able to design an algorithm which obtains logarithmic Neyman regret.

### 2.2. Related Works

The problem of off-policy evaluation, which generalizes ATE estimation, has been extensively studied in the literature on reinforcement learning (Dudík et al., 2011; Li et al., 2011; Jiang & Li, 2016). Most of the research in this area has focused on offline estimation, leading to precise characterizations of minimax lower bounds along with matching upper bounds (Li et al., 2015; Wang et al., 2017; Duan et al., 2020; Ma et al., 2021). Beyond policy evaluation, these methods have been extended to estimate other quantities, such as the cumulative distribution function of rewards (Huang et al., 2021; 2022). However, there has been limited exploration of adaptive versions of these methods. Some existing work includes Hanna et al. (2017), which focuses on off-policy learning, and Konyushova et al. (2021), which integrates offline off-policy evaluation techniques with online data acquisition to enhance sample efficiency in policy selection. However, these works are primarily empirical.

A related area of research concerns inference procedures for adaptively collected data. This can be categorized into asymptotic and non-asymptotic approaches. On the asymptotic side, one direction has focused on reweighting estimators and establishing their asymptotic normality (Hadad et al., 2021; Zhang et al., 2020; 2021). Another direction avoids asymptotics, instead leveraging modern advances in martingale theory to derive nonasymptotic confidence intervals and sequences for adaptively collected data, including estimates of the ATE (Howard et al., 2021; Waudby-Smith

& Ramdas, 2023; Waudby-Smith et al., 2022).

**Subsequent work.** After our initial arXiv posting, (Noarov et al., 2025) presented an extension of the `ClipOGD` algorithm to the fixed design setting, achieving logarithmic Neyman regret with respect to the optimal IPW estimator – which would imply linear regret against the stronger baseline we consider in this paper. Furthermore, this improved regret rate is notable, it requires substantially more restrictive assumptions than our analysis, including conditions that preclude standard stochastic reward models such as Bernoulli outcomes. These limitations reflect the inherent difficulty of obtaining logarithmic rates in their more challenging fixed design framework.

## 3. Background

**Problem Setup** We are interested in adaptive estimation of the average treatment effect. During each round, $t$, `Alg` uses the history of past observations $\mathcal{H}_{t-1} = \{(\pi_s, A_s, R_s)\}_{s=1}^{t-1}$ to select the probability of treatment allocation $\pi_t$. Then, $\pi_t$ is used to assign the next experimental unit to either the control ($A_t = 0$) or the treatment ($A_t = 1$) by sampling $A_t \sim \texttt{Bernoulli}(\pi_t)$. Finally, after assigning the experimental unit, we observe the outcome $R_t$ which marks the end of round $t$.

We formalize the above interaction protocol as follows. Let $\mathcal{F}_t = \sigma(\mathcal{H}_t)$ denote the filtration generated by the past observations. An algorithm $\texttt{Alg} = \{(\pi_t, h_t)\}_{t=1}^T$ is defined as a sequence of $\mathcal{F}_{t-1}$ measurable random elements where $\pi_t \in [0, 1]$ is the treatment allocation probability and $h_t : (\pi_t, A_t, R_t) \mapsto \mathbb{R}_{\geq 0}$ which can be thought of as the ATE estimate produced by `Alg` on round $t$.

We assume that the rewards are generated as $R_t = \mathbb{I}[A_t = 1] R_t(1) + \mathbb{I}[A_t = 0] R_t(0)$, where $R_t(a)$ are called the potential outcomes. We assume that the sequence of potential outcomes are jointly distributed according to some probability measure $\nu$ (the "environment") that satisfies the following assumptions. The first assumption is that the rewards are unconfounded, which means that, given $\mathcal{F}_{t-1}$, the potential outcomes $R_t(1), R_t(0)$ are conditionally independent of the treatment assignment $A_t$, i.e $R_t(1), R_t(0) \perp A_t \mid \mathcal{F}_{t-1}$. The second assumption is that the reward means and variances are conditionally fixed so that for all $t$, we have $\mathbb{E}_\nu[R_t(a) \mid \mathcal{F}_{t-1}] = r^\star(a)$ and $\mathbb{V}_\nu[R_t(a) \mid \mathcal{F}_{t-1}] = \sigma^2(a)$.

Our objective within this framework is to estimate the ATE $\Delta$, which is defined as

$$\Delta = r^\star(1) - r^\star(0).$$

**The `A2IPW` Estimator.** An algorithm for adaptive ATE estimation thus requires us to specify a method to compute the treatment allocation probability $\pi_t$ as well as the estimate $h_t$. A natural choice for $h_t$ is the AIPW estimator, which given some reward estimate $\widehat{r}$, is defined as

$$h_t = \frac{g(A_t)}{\mathbb{P}_{\texttt{Alg},\nu}(A_t)}(R_t - \widehat{r}(A_t)) + \widehat{\Delta}^{(\widehat{r})}, \quad (1)$$

where $g(A_t) = \mathbb{I}[A_t = 1] - \mathbb{I}[A_t = 0]$ and $\widehat{\Delta}^{(\widehat{r})} = \widehat{r}(1) - \widehat{r}(0)$. However, this estimator isn't well suited to sequential estimation, motivating Kato et al. (2020) to propose the Adaptive AIPW (`A2IPW`) estimator. Specifically, letting $\widehat{r}_t$ denote any $\mathcal{F}_{t-1}$ measurable function (i.e. a *predictable* reward estimate), they defined

$$h_t = \frac{\mathbb{I}[A_t = 1] - \mathbb{I}[A_t = 0]}{\pi_t(A_t)}(R_t - \widehat{r}_t(A_t)) + \widehat{\Delta}^{(\widehat{r}_t)}. \quad (2)$$

The `A2IPW` and AIPW estimators differ in two critical ways. First, the `A2IPW` estimator constructs propensity scores using the policy $\pi_t$ rather than the marginal probabilities $\mathbb{P}_{\texttt{Alg},\nu}(A_t)$, avoiding the computational infeasibility of computing marginal probabilities for adaptive sampling algorithms. Second, the `A2IPW` incorporates predictable rewards $\widehat{r}_t$, enabling sequential updates to reward estimates and fully adaptive algorithm design without phase-based approaches. Beyond these computational advantages, the `A2IPW` estimator possesses strong theoretical properties: it achieves asymptotic optimality necessary for sublinear Neyman regret, and recent developments have produced tight confidence sequences for this estimator, facilitating sequential testing and uncertainty quantification. These properties make the `A2IPW` estimator a natural choice for adaptive ATE estimation.

**Neyman Allocation and Regret** We use the mean squared error (MSE) to measure the quality of the estimates produced by our algorithm. However, by itself, the MSE is difficult to interpret because it does not consider the inherent difficulty of the problem. Therefore, we would like to normalize this error with respect to some problem-dependent baseline which we now define and motivate. Hahn et al. (2009) show that for any fixed allocation, $\pi$, the minimum attainable MSE of any estimator is

$$\frac{\sigma^2(1)}{\pi} + \frac{\sigma^2(0)}{1 - \pi}. \quad (3)$$

The Neyman allocation $\pi^\star$ is defined as the allocation which minimizes the above variance and a simple calculation shows that

$$\pi^\star = \frac{\sigma(1)}{\sigma(0) + \sigma(1)}. \quad (4)$$

Ideally, we would like to design an algorithm whose variance is close to this baseline and in order to understand the

rate at which this occurs, we consider the Neyman regret which is defined as

$$\mathfrak{R}_T = T \cdot \left( \texttt{MSE}\left(\widehat{\Delta}_T\right) - \texttt{MSE}^{\star} \right), \tag{5}$$

where $\texttt{MSE}^{\star}$ is the mean square error attained by an oracle algorithm which sets $\pi_t = \pi^{\star}$ for all $t \in \{1, \ldots, T\}$. The Neyman regret is simply the difference in the normalized MSE between the optimal variance and the MSE of the estimate produced by $\texttt{Alg}$. This normalization guarantees that the MSE converges to a constant (rather than 0), so that if $\texttt{Alg}$ has sublinear regret, then we are guaranteed that its MSE converges to the optimal MSE.

Using the fact that the $\texttt{A2IPW}$ is unbiased, along with the fact that $\pi_t$ and $\widehat{r}_t$ are predictable, we can express the Neyman regret as

$$\mathfrak{R}_T = \sum_{t=1}^{T} \mathbb{E}_{\texttt{Alg},\nu}\left[\ell(\pi_t, \widehat{r}_t)\right] - \ell(\pi^{\star}, r^{\star}), \tag{6}$$

where

$$\ell(\pi, r) = \sum_{a \in \{0,1\}} \frac{\sigma^2(a)}{\pi(a)} + \frac{1 - \pi(a)}{\pi(a)} \varepsilon_t^2(a) \tag{7}$$

is the Neyman loss and $\varepsilon_t(a) = r^{\star}(a) - \widehat{r}_t(a)$ is the reward estimation error.

**Notation.** In what follows, we will let

$$N_t(a) = \sum_{s=1}^{t} \mathbb{I}\left[A_s = a\right]$$

denote the number of times the action $a$ is selected at the end of round $t$,

$$\mu_t(a) = \frac{1}{N_t(a)} \sum_{s=1}^{t} R_s \mathbb{I}\left[A_s = a\right]$$

denote the empirical mean after $t$ rounds, and

$$\widehat{\sigma}_t^2(a) = \frac{1}{N_t(a)} \sum_{s=1}^{t} \left(R_s \mathbb{I}\left[A_s = a\right] - \mu_t(A)\right)^2$$

denote the emprical variance. We use $\widetilde{\mathcal{O}}(\cdot)$ to denote asymptotic equivalence up to doubly logarithmic factors.

## 4. The Optimistic Policy Tracking Algorithm

In this section, we introduce our Optimistic Policy Tracking (OPT) algorithm. We begin with a discussion of the difficulties of adaptive ATE estimation and the suboptimality of existing approaches. Next, we introduce our algorithm and provide insight into why it resolves the issues of existing approaches. Finally, we conclude with a brief discussion of the algorithmic design principles underlying our algorithm and their relation to ideas in the literature.

### 4.1. Preliminaries

**The difficulties of adaptive ATE estimation.** The primary difficulty of adaptive ATE estimation is in balancing the exploration-exploitation trade-off that arises from adaptive allocation. If we condition on $\mathcal{F}_{t-1}$ some algebra shows (see Lemma C.1) that the variance of the $\texttt{A2IPW}$ estimator is

$$\sum_a \frac{\sigma^2(a)}{\pi_t(a)} + \frac{1 - \pi_t(a)}{\pi_t(a)} \left(r^{\star}(a) - \widehat{r}_t(a)\right)^2, \tag{8}$$

which is minimized by setting $(\pi, r) = (\pi^{\star}, r^{\star})$ where $\pi^{\star}$ is the Neyman allocation introduced in Section 3. Since $\pi^{\star}$ and $r^{\star}$ are not known a priori, we need to design an algorithm to adaptively estimate them. However, this is challenging because optimizing the exploration allocation separately for estimating $\pi^{\star}$ and $r^{\star}$ (each requiring a different allocation) results in a procedure with high Neyman regret. As such, designing an algorithm to adaptively balance the exploration of $\pi^{\star}$ and $r^{\star}$ while simultaneously minimizing the Neyman regret becomes a very delicate task.

**Insights into improvements.** In order to better understand the improvements that can be made, we investigate previous approaches for balancing this trade-off. To simplify the exposition, in this section we assume that $\pi^{\star} \leq \frac{1}{2}$. The primary approach that past works (both asymptotic and nonasymptotic) have utilized is clipping the allocation. In fact, the algorithms proposed by Cook et al. (2024), Dai et al. (2023), Neopane et al. (2025) all utilize a clipping approach which computes the empirical allocation

$$\widehat{\pi}_t = \frac{\widehat{\sigma}_t(1)}{\widehat{\sigma}_t(0) + \widehat{\sigma}_t(1)},$$

and plays a clipped version of this estimate

$$\pi_t = \min\left\{1 - c_t, \max\left\{c_t, \widehat{\pi}_t\right\}\right\},$$

for some carefully chosen clipping sequence $c_t$ satisfying $c_t \to 0$. However, these clipping approaches have some important limitations.

The first limitation is that a clipping approach cannot be fully adaptive to the underlying problem instance because the clipping sequence must be chosen a priori. As such, past works choose $c_t$ in order to optimize the performance of their algorithm in a worst-case sense, leading to suboptimal Neyman regret for easy problem instances. As an example, Neopane et al. (2025) show that setting $c_t = t^{-\frac{1}{3}}$ is optimal when we are not willing to bound $\pi^{\star}$ away from 0 and 1. However, many practical problems are typically much easier than the worst case, and so we would like a procedure which is able to adapt to the underlying problem instance more appropriately.

The second, more pressing issue is that clipping approaches lead to algorithms which under-exploit, which is caused by

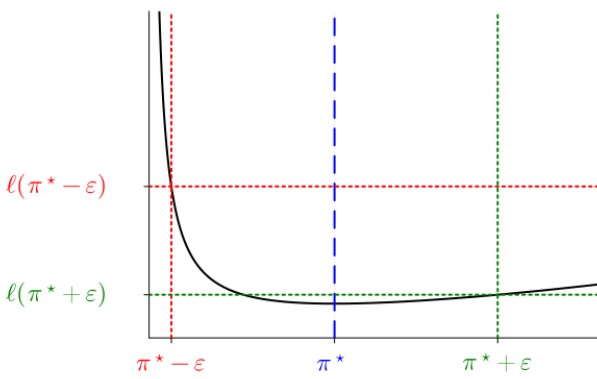

*Figure 1.* A plot of $\ell(\pi, r^\star)$ where $\pi^\star < \frac{1}{2}$. Note how the Neyman loss is smaller for $\pi^\star + \epsilon$. This is due to the fact that $\pi^\star + \epsilon$ is closer to $\frac{1}{2}$, highlighting how less exploratoratory allocations incur larger Neyman regret.

the asymmetry of the Neyman loss. To demonstrate this issue, in Figure 1, we plot the Neyman loss $\ell(\pi, r^\star)$ for a problem with $\pi^\star < \frac{1}{2}$. In this figure, we consider the Neyman loss at the points $\pi_+ = \pi^\star + \epsilon$ and $\pi_- = \pi^\star - \epsilon$. It is easy to see that $\ell(\pi_+, r^\star) < \ell(\pi_-, r^\star)$, and in fact this problem only worsens as $\pi^\star \to \{0, 1\}$. Practically, the implication is that an algorithms which under-sampled the arm with a smaller probability according to the Neyman allocation must necessarily pay a higher price than the same algorithm which over-sampled the same arm by the same amount. This is not merely a theoretical issue — we see in our experiments that while clipping-based approaches produce allocations which are closer to the Neyman allocation, they still have significantly worse empirical performance.

### 4.2. Optimistic Policy Tracking

**Main Algorithm.** Our proposed algorithm, OPT, is designed to address these aforementioned issues. Indeed, as we will see, not only does OPT better adapt to the underlying problem instances, it also better handles the exploration-exploitation trade-off when compared to prior works. The algorithm itself if simple and plays the allocation

$$\pi_t = \operatorname*{argmin}_{\pi \in \mathcal{CS}_t(\pi^\star)} \left| \frac{1}{2} - \pi \right|, \tag{9}$$

where $\mathcal{CS}_t(\pi^\star)$ is a confidence sequence for the Neyman allocation. For reward estimation, we simply use the sample mean $\widehat{r}_t(a) = \frac{1}{N_{t-1}(a)} \sum_{s=1}^{t-1} R_s \cdot \mathbb{I}[A_t = a]$.

The main difficulty now is in constructing the confidence sequence $\mathcal{CS}_t(\pi^\star)$. In order to do so, we first construct confidence sequences for the standard deviations of each arm. This is accomplished in Lemma B.1, which constructs a confidence sequence $\mathcal{CS}_t(\sigma(a)) = [\mathcal{L}_t(\sigma(a)), \mathcal{U}_t(\sigma(a))]$

whose with scales like $\mathcal{O}\left( \sqrt{\frac{\log\log t + \log \frac{1}{\delta}}{t}} \right)$. Using these confidence sequences on $\sigma(a)$, we can construct a confidence sequence for the Neyman allocation as follows

$$\mathcal{CS}_t(\pi^\star) = \left[ \frac{\mathcal{L}_t(\sigma(1))}{\mathcal{U}_t(\sigma(0)) + \mathcal{L}_t(\sigma(1))}, \frac{\mathcal{U}_t(\sigma(1))}{\mathcal{L}_t(\sigma(0)) + \mathcal{U}_t(\sigma(1))} \right]. \tag{10}$$

The full algorithm is provided in Algorithm 1.

---
**Algorithm 1** Optimistic Policy Tracking (`OPTrack`)
---
1: **for** $t = 1, 2, \ldots$ **do**
2:     Compute $\mathcal{CS}_t(\pi^\star)$ according to equation (10)
3:     Set $\pi_t = \operatorname{argmin}_{\pi \in \mathcal{CS}_t(\pi^\star)} \left| \frac{1}{2} - \pi \right|$
4:     Sample $A_t \sim \text{Bernoulli}(\pi_t)$
5:     Observe $R_t \sim \nu(A_t)$
6:     Compute $h_t$ according to equation (2)
7: **end for**
---

**Interpretation as Optimism.** We can interpret our algorithm as implementing the celebrated principle of *optimism in the face of uncertainty*. Optimism is an algorithmic design principle which is the basis of many well-known MAB and reinforcement learning algorithms (such as the "upper confidence bound"). Roughly speaking, the principle states that we should act as if the underlying problem instance is the easiest instance, which is feasible according to our past observations. In the regret minimization framework, this means playing the arm which has the largest upper confidence bound. For adaptive ATE estimation, this involves playing the allocation that is closest to $\frac{1}{2}$. This is because the difficulty of a problem is determined by the deviation of the Neyman allocation from $\frac{1}{2}$ – when the Neyman allocation is close to $\frac{1}{2}$, the objectives of exploration and exploitation are aligned. Suppose the Neyman allocation deviates from $\frac{1}{2}$, then as the allocation we play converges to the Neyman allocation, we are necessarily under-sampling one arm and thus *slowing* down our convergence to the Neyman allocation. This intuition is supported by the results of Neopane et al. (2025) and Dai et al. (2023) who show that the Neyman regret scales inversely with $\left| \pi - \frac{1}{2} \right|$. Therefore, implementing optimism for adaptive ATE estimation involves playing the most feasible allocation (as determined by our past observations) closest to $\frac{1}{2}$ – this is exactly the driving principle behind our `OPTrack` algorithm.

## 5. Results

In this section, we build our intuition on the behavior of `OPTrack` and conclude by stating our main result which is a bound on the Neyman regret of `OPTrack`.

Before we begin, we introduce some additional notation which will make our exposition easier. For any $\pi$, we define $\Delta(\pi) = \left|\frac{1}{2} - \pi\right|$ and $\underline{\pi} = \min\{\pi, 1 - \pi\}$. Additionally, we let $\Delta_{(\sigma)} = \sigma(1) - \sigma(0)$.

Our analysis begins with a Concentration result which demonstrates that the standard deviations concentrate at a 1 over square root of t rate. Specifically, in Lemma B.1, we demonstrate that with probability at least $1 - \delta$, for all $t \geq 2$ we have that

$$|\sigma - \widehat{\sigma}_t| \leq \mathcal{O}\left(\sqrt{\frac{\log\log t + \log \delta^{-1}}{t}}\right). \qquad (11)$$

This result follows through generalizing a similar result from (Audibert et al., 2006) utilizing modern martingale arguments (Howard et al., 2021). This enables us to obtain time uniform concentration which is valid in our setting with adaptively collected data.

To proceed, we split the behavior of OPTrack into two phases, an exploration *exploration phase* and the *concentration phase*. We define the exploration phase as the rounds for which $\pi_t = \frac{1}{2}$. During the early stages of interaction, we expect that each arm has been played sufficiently few times so that $\frac{1}{2} \in \mathcal{CS}_t(\pi^\star)$, and the exploration time $\mathbf{T}$ is the length of this phase. Intuitively, during this phase, there is not enough information in our observations to reliably predict $\pi^\star$ and so our best choice is to explore each arm uniformly. Fortunately, the length of this phase is not too long, and our first result bounds the length of this phase in terms of the absolute distance between the standard deviations.

**Lemma 5.1.** *Define the exploration time as*

$$\mathbf{T} = \min\left\{t : \pi_t \neq \frac{1}{2}\right\}. \qquad (12)$$

*Then, with probability at least $1 - \delta$, we have*

$$\mathbf{T} = \widetilde{\mathcal{O}}\left(\Delta_{(\sigma)}^{-2} \log \frac{1}{\delta}\right). \qquad (13)$$

The proof of this result is given in Appendix A.2. This result shows that OPTrack is able to adapt to the difficulty of the underlying problem instance — if the gap between the standard deviations is large, then the exploration phase will be short, and if the gap is small, then the exploration phase will be longer.

Once the exploration phase is over, the algorithm will be able to focus on the concentration phase. In this phase, optimism guarantees $\Delta(\pi_t) < \Delta(\pi^\star)$. Therefore, we can control the number of times each arm is played which we can in turn convert to bounds on $|\pi_t - \pi^\star|$.

Our next result formalizes this intuition.

**Lemma 5.2.** *With probability at least $1 - \delta$, we have that*

$$\pi_t - \pi^\star = \widetilde{\mathcal{O}}\left(\sqrt{\frac{\log \frac{1}{\delta}}{\underline{\pi}^\star \cdot t}} \cdot \frac{1}{\sigma(0) + \sigma(1)}\right). \qquad (14)$$

The reason for the appearance of $\underline{\pi}^\star$ is due to the convergence of $\pi_t$ based on the number of times that both arms have been played. If we play one arm too often, then the width of the confidence interval for $\pi^\star$ would depend entirely on the width of the lesser sampled arm.

Our main result combines the above lemmas to provide a bound on the Neyman regret.

**Theorem 5.3.** *With probability at least $1 - \delta$, the Neyman regret of OPTrack is upper-bounded as*

$$\widetilde{\mathcal{O}}\left(\Delta_{(\sigma)}^{-2} + \left(\frac{1}{\underline{\pi}^\star}\right)^2 \log T\right). \qquad (15)$$

The first term above is the per-round Neyman regret during the exploration phase and our bound follows from the fact that the Neyman regret is at most 4 when we play $\pi_t = \frac{1}{2}$. The second term in our bound is the Neyman regret during the concentration phase and follows from the application of Lemma 5.2 in conjunction with Lemma 2 of (Neopane et al., 2025) showing that the Neyman regret scales according to $|\pi^\star - \pi_t|^2 \approx \frac{1}{\pi^\star \cdot t}$. Since the contribution to the Neyman regret from the reward estimation also scales like $\frac{1}{\pi^\star \cdot t}$, taking a sum over these two terms gives us the desired result.

In order to get a better understanding of our result, we consider the behavior of a hypothetical algorithm which plays the optimal Neyman allocation $\pi^\star$ but incurs a loss based on the empirically computed allocation, $\pi_t$. A simple calculation shows that $\pi_t$ converges to $\pi^\star$ at a rate of $\Theta\left((\pi^\star \cdot t)^{-\frac{1}{2}}\right)$. This in turn implies that the Neyman regret would be

$$\widetilde{\mathcal{O}}\left(\left(\frac{1}{\underline{\pi}^\star}\right)^2 \log T\right), \qquad (16)$$

which, modulo the regret from the clipping phase, is the same as the Neyman regret incurred by OPTrack. This suggests that our algorithm is correctly adapting to the difficulty of the problem.

**Comparison with ClipSMT.** At first glance, our result appears to be quite similar to the Neyman regret bound from (Neopane et al., 2025) who similarly show a logarithmic bound on the Neyman regret. However, this is not the case, due to differing definitions of the Neyman regret. In (Neopane et al., 2025), the Neyman regret is defined with respect to the minimum variance over allocations for the fixed IPW estimator. Our Neyman regret is defined with

respect to the minimum attainable variance over any *pair* of estimators and allocations. This means that while our regret bounds share a similar form, the performance of our algorithm is significantly better than the performance of the `ClipSMT` algorithm. Concretely, using our definition of the Neyman regret to characterize the performance of the `ClipSMT` algorithm (as well as the `ClipOGD` algorithm), we see that these algorithms actually have *linear* Neyman regret since the variance of their policies cannot converge to the minimum attainable variance.

## 6. Experiments

In this section, we present experiments[1] to evaluate the empirical performance of our algorithm. We compare `OPTrack` against the `ClipSDT` algorithm proposed by Cook et al. (2024), as well as two oracle algorithms that follow the Neyman allocation. One of these oracle algorithms sequentially estimates the reward, while the other has access to the true reward. We evaluate these algoritmhs on a range of synthetic simulations as well as on a macro-insurance intervention dataset (Groh & McKenzie, 2016).

We do not include results for the `ClipSMT` and `ClipOGD` algorithms, as their variances fail to converge to the oracle variance, consistently leading to significantly worse performance than the other algorithms which obscures the clarity of the plots. This outcome is expected, given that both algorithms incur linear Neyman regret.

We consider 6 problem instances where both arms follows Bernoulli distributions. For each of these problem instances, we fix the treatment mean to be $\frac{1}{2}$ and vary $\mu_0 \in \{0.05, 0.1, 0.2, 0.3, 0.4, 0.5\}$ which in turn leads to problem instances with different Neyman allocations.

For each of these problems, we run `OPTrack`, `ClipSDT`, and the reward estimation oracle for $T \in \{100, 200, 300, 400, 500, 600, 700, 800, 900, 1000, 1500, 2000\}$ and plot the normalized MSE ($T \cdot \text{MSE}$) over 500,000 simulations. For the oracle baseline, we explicitly compute the MSE. The results of these simulations are given in Figure 2. We include error bars in Figure 2, though they are barely visible due to the large number of simulations.

Our results show that `OPTrack` consistently outperforms `ClipSDT` over all problem instances. The difference between the two becomes negligible for larger values of $T$ which is expected since all algorithms eventually converge to the Neyman allocation and true reward function. However, for smaller sample sizes, we see that `OPTrack` provides around a 10-15 percent improvement over `ClipSDT`. This improvement is due to the reasons given in Section 4.

---

[1]Code for replicating experiments can be found at the following GitHub repo: https://github.com/oneopane/adaptive-ate-estimation.

The performance of `OPTrack` is competitive with the reward estimation oracle for moderate values of $\pi^\star$ and even outperforms the reward estimation oracle on some problem instances. This is because `OPTrack` is more exploratory and obtains better reward estimates early on.

We additionally perform experiments on a macro-insurance intervention dataset (Groh & McKenzie, 2016) which investigates the effects of macro-insurance in Egypt. The results from this experiment are given in Table 1. Our experiments qualitatively align with the results from our synthetic experiments, demonstrating that our proposed algorithm outperforms prior approaches, especially in the small sample regime.

| Algorithm | 100 | 300 | 500 | 1000 | 1500 |
|---|---|---|---|---|---|
| ClipSDT | 0.117 | 0.102 | 0.098 | 0.093 | 0.095 |
| OPTrack | 0.103 | 0.095 | 0.094 | 0.093 | 0.092 |
| Est. Reward Oracle | 0.10 | 0.097 | 0.094 | 0.093 | 0.094 |
| Oracle | 0.092 | 0.090 | 0.093 | 0.090 | 0.091 |

*Table 1.* Mean square error of the `ClipSDT`, `OPTrack` algorithms as well as two oracle baselines – an oracle which has knowledge of $\pi^\star$, and an oractle with knowlege of $\pi^\star$ and $r^\star$ on a macro-insurance intervention dataset. Note that `ClipSDT` is competitive with the Oracle, even in the small sample regime.

## 7. Conclusion

This work proposed a new algorithm for adaptive ATE estimation. We identified some key issues with past approaches which limited their performance both empirically and theoretically and demonstrated how to resolve them. Our proposed solution borrows ideas from the literature on Regret Minimization and showed how to extend some of these ideas to the problem of adaptive ATE estimation. We believe that these ideas will be crucial for developing adaptive algorithms for inference for more complicated settings as well as for related problems like Off-Policy Evaluation.

### 7.1. Future Work

We believe there are a few directions for future work that we find very compelling. The first is the extension of our algorithm to the setting with covariates and with more sophisticated reward estimation. In the causal inference literature, practitioners typically use nonparametric regression to estimate the $r^\star$ and so extending our ideas to work with such estimators warrants more attention. Another interesting direction is the extension to multiple arms. Here we believe that the correct extension is to compute a confidence interval around the Neyman allocation, and then project this set onto the Uniform distribution over the actions. The primary difficulty for this extension is in the analysis – if we apply our techniques directly, this will result in an additional

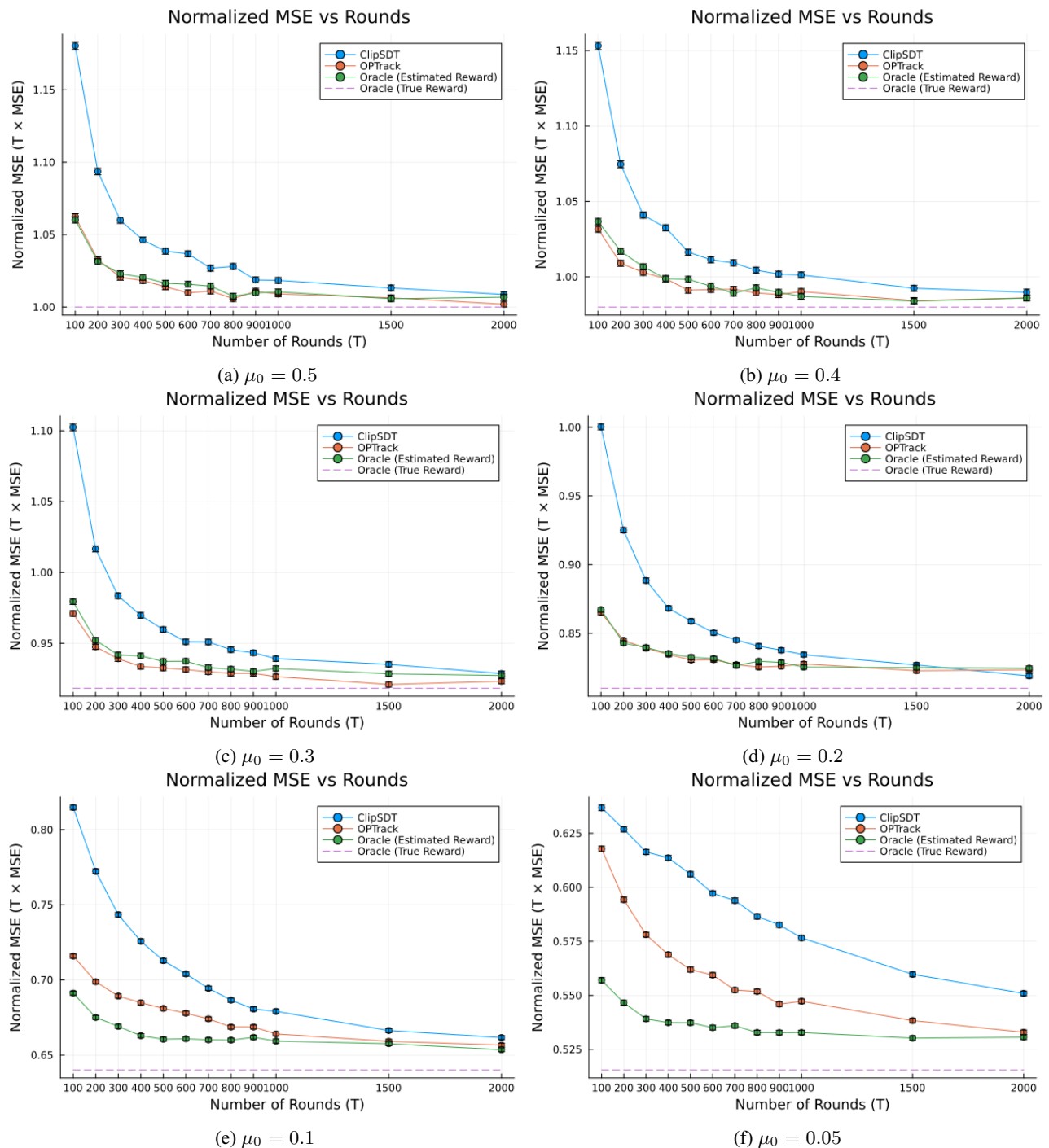

*Figure 2.* Normalized MSE ($T \cdot$MSE) for `OPTrack`, `ClipSDT`, and the oracle baselines across six problem instances, each with Bernoulli rewards with $\mu_1 = \frac{1}{2}$ and $\mu_0 \in \{0.05, 0.1, 0.2, 0.3, 0.4, 0.5\}$. Results are averaged over 500,000 simulations. `OPTrack` consistently outperforms `ClipSDT`, with a 10-15 improvement for smaller $T$. Notably, `OPTrack` is competitive with the reward estimation oracle and even outperforms it in some cases due to better exploration of the reward function early on. As $T$ increases, all algorithms converge to the oracle baseline.

factor of $K$ in the term that is dependent on $T$, where $K$ is the number of arms. It is an interesting question to see if our analysis can be improved to remove this additional factor.

Finally extending these ideas to more complicated interaction protocols such as Reinforcement Learning warrants further study.

## Acknowledgments

This work is sponsored in part by AFRL Center of Excellence grant FA86501925209 and NSF AI Institute for Societal Decision Making under grant IIS2229881.

## Impact Statement

While our paper is primarily theoretical, we believe that the insights developed will be important for downstream applications such as causal inference which has broad applications over a variety of fields including clinical trials and A/B testing.

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

## A. Analysis of Optimistic Policy Tracking

**Preliminaries** We will begin by defining our good event. Consider the following events

$$\mathcal{E}_\sigma(\delta) = \bigcap_{a,t \in \mathbb{N}} \left\{ |\widehat{\sigma}_t(a) - \sigma(a)| \le 4.2 \sqrt{\frac{\ell(t,\delta)}{t}} \right\} \tag{17}$$

$$\mathcal{E}_N(\delta) = \bigcap_{t \in \mathbb{N}} \left\{ \left| N_t(a) - \sum \pi_t(a) \right| \le \sqrt{t\ell(t,\delta)} \right\} \tag{18}$$

$$\mathcal{E}_r(\delta) = \bigcap_{at \in \mathbb{N}} \left\{ |\widehat{r}_t(a) - r^\star(a)| \le \sqrt{t\ell(t,\delta)} \right\}. \tag{19}$$

Let $\tilde{\delta} = \frac{\delta}{5}$ and define the good event $\mathcal{E}(\tilde{\delta}) = \mathcal{E}_\sigma(\delta) \cap \mathcal{E}_N(\delta) \cap \mathcal{E}_r(\delta)$. Applying Lemma B.1 to control $\mathcal{E}_\sigma(\tilde{\delta})$ and Theorem 1 from (Howard et al., 2021) to control $\mathcal{E}_N(\tilde{\delta})$, and $\mathcal{E}_r(\tilde{\delta})$ shows that the event $\mathcal{E}(\tilde{\delta})$ occurs with probability at least $1 - \delta$. Throughout the remained of this section, we assume the good event holds.

### A.1. Proof of Theorem 1

We begin by decomposing the Neyman regret

$$\mathfrak{R}_T = \sum_{t=1}^T \ell(\pi_t, \widehat{r}_t) \tag{20}$$

$$= \sum_{t=1}^T \sum_a \left( \frac{\sigma^2(a)}{\pi_t(a)} + \frac{1 - \pi_t(a)}{\pi_t(a)} \varepsilon_t^2(a) - \frac{\sigma^2(a)}{\pi^\star[a]} \right) \tag{21}$$

$$= \sum_{t=1}^{\mathbf{T}} \sum_a \left( \frac{\sigma^2(a)}{\pi_t(a)} + \frac{1 - \pi_t(a)}{\pi_t(a)} \varepsilon_t^2(a) - \frac{\sigma^2(a)}{\pi^\star[a]} \right) + \sum_{t=\mathbf{T}+1}^T \sum_a \left( \frac{\sigma^2(a)}{\pi_t(a)} + \frac{1 - \pi_t(a)}{\pi_t(a)} \varepsilon_t^2(a) - \frac{\sigma^2(a)}{\pi^\star[a]} \right). \tag{22}$$

For the first term, we have that $\pi_t = \frac{1}{2}$, and $\varepsilon_t(a) \le 1$, so that

$$\sum_a \left( \frac{\sigma^2(a)}{\pi_t(a)} + \frac{1 - \pi_t(a)}{\pi_t(a)} \varepsilon_t^2(a) - \frac{\sigma^2(a)}{\pi^\star[a]} \right) \tag{23}$$

$$\le \sum_a \left( \frac{\sigma^2(a)}{\pi_t(a)} - \frac{\sigma^2(a)}{\pi^\star[a]} \right) + 2 \tag{24}$$

$$\le 4, \tag{25}$$

to so that the regret from the exploration phase is $4\mathbf{T}$.

For the second term, we have

$$\sum_a \left( \frac{\sigma^2(a)}{\pi_t(a)} + \frac{1 - \pi_t(a)}{\pi_t(a)} \varepsilon_t^2(a) - \frac{\sigma^2(a)}{\pi^\star[a]} \right) \tag{26}$$

$$= \sum_a \left( \frac{\sigma^2(a)}{\pi_t(a)} - \frac{\sigma^2(a)}{\pi^\star(a)} \right) + \sum_a \left( \frac{1 - \pi_t(a)}{\pi_t(a)} \varepsilon_t^2(a) \right) \tag{27}$$

$$\tag{28}$$

We can bound the first term by applying Lemma 4.3 from (Neopane et al., 2025) in conjunction with so that

$$\sum_a \left( \frac{\sigma^2(a)}{\pi_t(a)} - \frac{\sigma^2(a)}{\pi^\star(a)} \right) \le \frac{625}{(\sigma(0) + \sigma(1))^2} \frac{\ell(t,\delta)}{\pi^\star t} \tag{29}$$

In order to bound the second term, we observe that on the good event

$$|r^\star(a) - \widehat{r}_t(a)| \leq \sqrt{\frac{\ell(t,\delta)}{N_t(a)}} \tag{30}$$

$$\leq \sqrt{\frac{\ell(t,\delta)}{\pi^\star t - \sqrt{t\ell(t,\delta)}}} \tag{31}$$

$$\leq 2\sqrt{\frac{\ell(t,\delta)}{\pi^\star t}}, \tag{32}$$

where in the last line we have again applied Lemma 4.5 from (Neopane et al., 2025).

Therefore, we have that

$$\sum_a \left(\frac{1 - \pi_t(a)}{\pi_t(a)} \varepsilon_t^2(a)\right) \leq \frac{8\ell(t,\delta)}{(\pi^\star)^2 t} \tag{33}$$

We can bound the sum of these two terms as $625\frac{\ell(t,\delta)}{(\pi^\star)^2 t}$. The result then follows by summing this over $t < T$ and adding the Neyman regret from the exploration phase.

### A.2. Proof of Lemma 5.1

*Proof.* Suppose, without loss of generality, that $\pi^\star < \frac{1}{2}$; in order to obtain results for $\pi^\star > \frac{1}{2}$, we can simply flip the roles of the treatment and control arms. For the case that $\pi^\star = \frac{1}{2}$, then `OPTrack` will always play $\pi_t$.

Since $\pi^\star < \frac{1}{2}$, bounding $\mathbf{T}$ is equivalent to determining the largest time $t$ such that $\mathcal{U}_t(\pi^\star) < \frac{1}{2}$, i.e we wish to compute

$$\min\left\{t : \frac{\sigma(1) + 4.2\sqrt{\frac{\ell(t,\delta)}{N_t(1)}}}{\sigma(0) + \sigma(1) + 4.2\sqrt{\frac{\ell(t,\delta)}{N_t(1)}} - 4.2\sqrt{\frac{\ell(t,\delta)}{N_t(0)}}} < \frac{1}{2}\right\} \tag{34}$$

Using the fact that $\pi_t = \frac{1}{2}$ for all $t < \mathbf{T}$, can control

$$N_t(a) \in \left[\frac{t}{2} \pm 1.7\sqrt{t\ell(t,\delta)}\right]. \tag{35}$$

Plugging this into equation (34) and rearranging shows that we need to bound

$$\min\left\{t : \frac{\ell(t,\delta)}{t\left(\frac{1}{2} - 1.7\sqrt{\frac{\ell(t,\delta)}{t}}\right)} < \frac{\Delta_{(\sigma)}^2}{18}\right\}. \tag{36}$$

Applying Lemma B.10 from (Neopane et al., 2025) shows that whenever $t \geq \widetilde{\mathcal{O}}\left(\log(\frac{1}{\delta})\right)$, we have that $1.7\sqrt{\frac{\ell(t,\delta)}{t}} < \frac{1}{4}$ so that we need to bound

$$\min\left\{t : t > \frac{64}{\Delta_{(\sigma)}^2}\ell(t,\delta)\right\}. \tag{37}$$

Another application of Lemma B.10 shows that this quantity is bounded by

$$\frac{64}{\Delta_{(\sigma)}^2}\log\frac{5.2}{\delta} + \frac{64}{\Delta_{(\sigma)}^2}\log\log\frac{64}{\Delta_{(\sigma)}^2} \tag{38}$$

which gives us the desired result.

$\square$

## A.3. Proof of Lemma 5.2

**Lemma A.1.** *Let $t \geq \mathbf{T}$. Then, with probability at least $1 - \delta$, we have that*

$$\pi_{t+1} - \pi^\star \leq \frac{25}{\sigma(0) + \sigma(1)} \sqrt{\frac{\ell(t, \delta)}{\pi^\star t}}. \tag{39}$$

*Proof.* Wlog we assume $\pi^\star < \frac{1}{2}$ so that $\underline{\pi^\star} = \pi^\star$. First note that $s \geq \mathbf{T}$, we have that

$$\pi_{t+1} \in \left[\pi^\star, \frac{\sigma(1) + Z_{1,t}}{\sigma(0) + \sigma(1) + Z_{1,t} - Z_{0,t}}\right] \tag{40}$$

$$= \left[\pi^\star, \pi^\star \frac{\sigma(0) + \sigma(1)}{\sigma(0) + \sigma(1) + Z_{1,t} - Z_{0,t}} + \frac{Z_{1,t}}{\sigma(0) + \sigma(1) + Z_{1,t} - Z_{0,t}}\right] \tag{41}$$

$$\subset \left[\pi^\star, \frac{1}{2}\right], \tag{42}$$

where we have defined

$$Z_t(a) = 4.2\sqrt{\frac{\ell(t, \delta)}{N_t(a)}},$$

and equation (42) follows from the definition of the $\mathbf{T}$.

Since $\pi_t \in \left[\pi^\star, \frac{1}{2}\right]$, we know that $1 - \pi_t \in \left[\frac{1}{2}, 1 - \pi^\star\right]$ which we use to control the number of times each arm is played.

$$N_t(1) \geq \pi^\star \cdot t - \sqrt{t\ell(t, \delta)} \tag{43}$$

$$N_t(0) \geq \frac{t}{2} - \sqrt{t\ell(t, \delta)}. \tag{44}$$

Plugging these values into the upper bound in equation (41), some algebra shows that

$$\pi_{t+1} - \pi^\star = \pi^\star \frac{\sigma(0) + \sigma(1)}{\sigma(0) + \sigma(1) + Z_{1,t} - Z_{0,t}} + \frac{Z_{1,t}}{\sigma(0) + \sigma(1) + Z_{1,t} - Z_{0,t}} - \pi^\star \tag{45}$$

$$= \pi^\star \cdot \frac{Z_0(t) - Z_1(t)}{\sigma(0) + \sigma(1) + Z_{1,t} - Z_{0,t}} + \frac{Z_{1,t}}{\sigma(0) + \sigma(1) + Z_{1,t} - Z_{0,t}} \tag{46}$$

$$\leq \frac{Z_{0,t}}{\sigma(0) + \sigma(1) + Z_{1,t} - Z_{0,t}} + \frac{Z_{1,t}}{\sigma(0) + \sigma(1) + Z_{1,t} - Z_{0,t}} \tag{47}$$

$$\leq 8.4\sqrt{\frac{\ell(t, \delta)}{\pi^\star t - \sqrt{t\ell(t, \delta)}}} \cdot \left(\frac{1}{\sigma(0) + \sigma(1) - Z_{0,t}}\right). \tag{48}$$

Applying Lemma B.10 from Neopane et al. (2025), we have that when $t = \widetilde{\mathcal{O}}\left(\left(\frac{1}{\pi^\star}\right)^2 \log \frac{1}{\delta}\right)$, we have that $\pi^\star t - \sqrt{t\ell(t, \delta)} \geq \frac{1}{2}\pi^\star t$. Next, since $t \geq \mathbf{T}$, we have that

$$Z_{0,t} = 4.2\sqrt{\frac{\ell(t, \delta)}{t}} \tag{49}$$

$$\leq \frac{\Delta_{(\sigma)}}{8}. \tag{50}$$

Therefore,

$$\sigma(0) + \sigma(1) + Z_{1,t} \geq \sigma(0) + \sigma(1) - \frac{\Delta_{(\sigma)}}{8} \tag{51}$$

$$= \sigma(0) + \sigma(1) - \frac{\sigma(0) - \sigma(1)}{8} \tag{52}$$

$$\geq \frac{\sigma(0) + \sigma(1)}{2}. \tag{53}$$

Combining these results, we have that

$$\pi_{t+1} - \pi^\star \leq \frac{25}{\sigma(0) + \sigma(1)} \sqrt{\frac{\ell(t, \delta)}{\pi^\star t}}, \tag{54}$$

which proves the desired result. $\qquad\square$

## B. Concentration Results

The proof of this lemma is based on a similar proof found in (Audibert et al., 2006) and extends the results to hold in the sequential setting.

**Lemma B.1.** *Let $(X_t)$ be a $[0, 1]$-valued stochastic process defined on some filtration $(\mathcal{F}_t)$ satisfying $\mu = \mathbb{E}_{t-1}[X_t]$ and $\sigma^2 = \mathbb{V}_{t-1}[X_t]$. Define*

$$\mu_t = \frac{1}{t} \sum_{t=1}^{t} X_t \tag{55}$$

$$\widehat{\sigma}_t^2 = \frac{1}{t} \sum_{s=1}^{t} (X_t - \mu_t)^2. \tag{56}$$

*Then, with probability at least $1 - \delta$, for all $t \geq 2$ we have that*

$$\sigma \in \left[ \widehat{\sigma}_t - 1.7 \sqrt{\frac{\ell(t, \delta)}{t}}, \widehat{\sigma}_t + 4.2 \sqrt{\frac{\ell(t, \delta)}{t}} \right]. \tag{57}$$

*Proof.* Define $Y_t = (X_t - \mu)^2 - \sigma^2$, and $S_t = \sum_{i=1}^{t} Y_t$. Letting $\mathcal{V} = \mathbb{V}_{t-1}[Y_t]$, we apply Theorem 1 from (Howard et al., 2021) which gives us the following time-uniform Bernstein inequality (see Table 3 in the Appendix). Applying a union bound, we have with probability at least $1 - \delta$, for all $t \in \mathbb{N}$, that

$$|\mu_t - \mu| \leq 1.7\sigma \sqrt{\frac{\ell\left(t, \frac{\delta}{4}\right)}{t}} + 1.7 \frac{\ell\left(t, \frac{\delta}{4}\right)}{t}, \tag{58}$$

$$|Y_t| \leq 1.7 \sqrt{\frac{\mathcal{V}\ell\left(t, \frac{\delta}{4}\right)}{t}} + 1.7 \frac{\ell\left(t, \frac{\delta}{4}\right)}{4t} \tag{59}$$

$$\leq 1.7\sigma \sqrt{\frac{\ell\left(t, \frac{\delta}{4}\right)}{t}} + 1.7 \frac{\ell\left(t, \frac{\delta}{4}\right)}{t}, \tag{60}$$

where we set $\ell(t, \delta) = \log \log 2t + 0.72 \log \frac{5.2}{\delta}$ and the last inequality follows from the fact that $\mathcal{V} < \sigma^2$. Letting $\mu_t = \frac{1}{t} \sum_{s=1}^{t} X_s$ some algebra demonstrates that

$$
\begin{aligned}
S_t &= \sum_{i=1}^{t} (X_i - \mu)^2 - \sigma^2 \\
&= \sum_{i=1}^{t} \left[ ((X_i - \mu_t) - (\mu_t - \mu))^2 - \sigma^2 \right] \\
&= \sum_{i=1}^{t} \left[ (X_i - \mu_t)^2 + 2(X_i - \mu_t)(\mu_t - \mu) + (\mu_t - \mu)^2 - \sigma^2 \right] \\
&= t\sigma_t^2 + 2(\mu_t - \mu) \sum_{i=1}^{t} (X_i - \mu_t) + t(\mu_t - \mu)^2 - t\sigma^2 \\
&= t\sigma_t^2 + 0 + t(\mu_t - \mu)^2 - t\sigma^2 \\
&= t(\sigma_t^2 - \sigma^2 + (\mu_t - \mu)^2),
\end{aligned}
$$

which implies

$$\left(\sigma_t^2 - \sigma^2\right) = \frac{1}{t}\sum_{s=1}^{t} Y_s - (\mu_t - \mu)^2 \leq \frac{1}{t}\sum_{s=1}^{t} Y_s. \tag{61}$$

Letting $L = \frac{\ell(t,\delta)}{t}$, and applying the bounds in equations (58) and 60, some algebra shows that

$$\sigma^2 + 1.7\sigma\sqrt{L} + 1.7L - \sigma_t^2 \geq 0. \tag{62}$$

Completing the square and rearranging shows that

$$\sigma \geq \sqrt{\sigma_t^2 + \left(1.7^2 - 1.7\right)L} - 1.7\sqrt{L} \tag{63}$$

$$\geq \sigma_t - 1.7\sqrt{L}. \tag{64}$$

Repeating the same argument with $-Y_t$ shows that

$$\sigma \leq \sigma_t + 4.2\sqrt{L}. \tag{65}$$

Combining these bounds we have with probability at least $1 - \delta$, for all $t > 2$

$$\sigma \in \left[\sigma_t - 1.7\sqrt{\frac{\ell(t,\delta)}{t}}, \sigma_t + 4.2\sqrt{\frac{\ell(t,\delta)}{t}}\right]. \tag{66}$$

$\square$

## C. Misc. Results

**Lemma C.1.** *For any* `Alg`*, we have that*

$$\mathbb{V}_{Alg,\nu}\left[\widehat{\Delta}_T\right] = \frac{1}{T^2}\sum_{t=1}^{T} \mathbb{V}_{Alg,\nu}\left[\mathtt{AIPW}_t\right] \tag{67}$$

$$= \frac{1}{T^2}\sum_{t=1}^{T} \mathbb{E}_{Alg,\nu}\left[\sum_{a} \frac{\sigma^2(a)}{\pi_t(a)} + \left(\frac{1 - \pi_t(a)}{\pi_t(a)}\right)\varepsilon_{t-1}^2(a)\right] \tag{68}$$

*Proof.* Leting $z_t = \mathtt{AIPW}_t - \Delta$, we have

$$\mathbb{V}_{Alg,\nu}\left[\widehat{\Delta}_T\right] = \frac{1}{T^2}\mathbb{E}\left[\left(\sum_{t=1}^{T} z_t\right)^2\right] \tag{69}$$

$$= \frac{1}{T^2}\left(\sum_{t=1}^{T} \mathbb{E}\left[z_t^2\right] + \sum_{t=1}^{T}\sum_{s=1}^{t=1} \mathbb{E}\left[z_t \cdot z_s\right]\right) \tag{70}$$

$$= \frac{1}{T^2}\sum_{t=1}^{T} \mathbb{E}\left[z_t^2\right] \tag{71}$$

$$= \frac{1}{T^2}\sum_{t=1}^{T} \mathbb{V}\left[\mathtt{AIPW}_t\right]. \tag{72}$$

The applying the law of total variance shows that $\mathbb{V}_{Alg,\nu}\left[\mathtt{AIPW}_t\right] = \mathbb{E}_{Alg,\nu}\left[\mathbb{V}\left[\mathtt{AIPW}_t \mid \mathcal{F}_{t-1}\right]\right]$ since

$\mathbb{V}\left[\mathbb{E}\left[\texttt{AIPW}_t \mid \mathcal{F}_{t-1}\right]\right] = 0$. Computing the conditional variance, we obtain

$$\mathbb{V}_{\texttt{Alg},\nu}\left[\texttt{AIPW}_t \mid \mathcal{F}_{t-1}\right] = \mathbb{E}_{\texttt{Alg},\nu}\left[\left(\texttt{AIPW}_t - \Delta\right)^2 \mid \mathcal{F}_{t-1}\right] \tag{73}$$

$$= \mathbb{E}_{\texttt{Alg},\nu}\left[\left(w_t\left(\delta_t + \varepsilon_{t-1}\right) + \widehat{\Delta}_{t-1}^{(r)} - \Delta\right)^2 \mid \mathcal{F}_{t-1}\right] \tag{74}$$

$$= \mathbb{E}_{\pi_t}\left[w_t^2\left(\sigma^2 + \varepsilon_{t-1}^2\right) - \left(\Delta - \widehat{\Delta}_{t-1}^{(r)}\right)^2\right] \tag{75}$$

$$= \sum_a \frac{\left(\sigma^2(a) + \varepsilon_{t-1}^2(a)\right)}{\pi_t(a)} - \left(\varepsilon_{t-1}(1) - \varepsilon_{t-1}(0)\right)^2 \tag{76}$$

$$= \sum_a \left[\frac{\sigma^2(a)}{\pi_t(a)} + \left(\frac{1}{\pi_t(a)} - 1\right) \cdot \varepsilon_{t-1}^2(a)\right] + 2\varepsilon_{t-1}(1) \cdot \varepsilon_{t-1}(0) \tag{77}$$

$$= \sum_a \left[\frac{\sigma^2(a)}{\pi_t(a)} + \left(\frac{1 - \pi_t(a)}{\pi_t(a)}\right) \cdot \varepsilon_{t-1}^2(a)\right] + 2\varepsilon_{t-1}(1) \cdot \varepsilon_{t-1}(0). \tag{78}$$

Therefore, we have

$$\mathbb{V}_{\texttt{Alg},\nu}\left[\widehat{\Delta}_T\right] = \frac{1}{T^2}\sum_{t=1}^{T}\mathbb{E}\left[\sum_a\left(\frac{\sigma^2(a)}{\pi_t(a)} + \left(\frac{1 - \pi_t(a)}{\pi_t(a)}\right) \cdot \varepsilon_{t-1}^2(a)\right) + 2\varepsilon_{t-1}(1) \cdot \varepsilon_{t-1}(0)\right] \tag{79}$$

$$= \mathbb{E}_{\texttt{Alg},\nu}\left[\sum_a \frac{\sigma^2(a)}{\pi_t(a)} + \left(\frac{1 - \pi_t(a)}{\pi_t(a)}\right) \cdot \varepsilon_{t-1}^2(a)\right], \tag{80}$$

where the second inequality follows from the fact that $\varepsilon_t^2(a)$ are uncorrelated. $\qquad\square$

