# OpenReview forum: "Optimistic Algorithms for Adaptive Estimation of the Average Treatment Effect"
_ICML.cc/2025/Conference — ICML 2025 poster_

### Official Review · Reviewer_Gq4L · 2025-03-11

**Overall Recommendation:** 4

**Summary:**

This paper addresses the challenges of the exploration-exploitation tradeoff in the adaptive estimation of the Average Treatment Effect (ATE), proposing a novel approach based on Neyman regret. The authors clearly outline the motivation and theoretical foundation of the algorithm, and conduct simple experiments on synthetic datasets.

**Claims And Evidence:**

The authors provide a comprehensive and clear explanation of the motivation and theoretical justification for the proposed method. However, their second claimed contribution, the experimental validation of the method's effectiveness, seems somewhat overstated, as the experiments presented in the paper are overly simplistic and lack detailed descriptions of the experimental setup.

**Essential References Not Discussed:**

The relevant literature that needed to be cited has been correctly referenced.

**Experimental Designs Or Analyses:**

The experimental part of the paper is incomplete, which constitutes its major weakness.

1. Details of the simulations are missing, and the authors should provide them, such as the data generation process.

2. As mentioned in the related work section, there are many related works beyond ClipSDT [1], such as [2, 3]. It would be beneficial to compare the proposed method with some of the many related works, if possible.

3. It would also be helpful to include experiments on real-world datasets to validate the effectiveness of the method in real-world scenarios.

[1] Cook, Thomas, Alan Mishler, and Aaditya Ramdas. "Semiparametric efficient inference in adaptive experiments." Causal Learning and Reasoning. PMLR, 2024.

[2] Kato, Masahiro, et al. "Efficient adaptive experimental design for average treatment effect estimation." arXiv preprint arXiv:2002.05308 (2020).

[3] Kato, M., et al. Active Adaptive Experimental Design for Treatment Effect Estimation with Covariate Choice. International Conference on Machine Learning, PMLR, 2024.

**Methods And Evaluation Criteria:**

The proposed method is correct, supported by thorough theoretical proofs and illustrative examples. However, the paper lacks details on evaluation criteria, such as the data generation process of the simulations. Additionally, it would be beneficial to include experiments on real-world datasets to validate the effectiveness of the proposed method in real-world application scenarios.

**Other Comments Or Suggestions:**

The following are my comments after reviewing the rebuttal:

The majority of the questions and concerns have been well addressed:

1. The authors did not compare additional baseline methods because ClipSDT is a state-of-the-art method. As a result, they chose to omit baselines that have been shown to perform worse than ClipSDT, as well as those that do not converge at all.

2. The authors have provided a detailed description of their experimental setup on the synthetic dataset and have committed to including the relevant descriptions in the revised version.

3. The authors have added experiments on a real-world dataset, and the results show that the proposed OPTrack method outperforms the baselines, with performance approaching that of the oracle algorithm using the true reward.

**Other Strengths And Weaknesses:**

The strengths of this paper lie in its clear writing, well-defined and reasonable motivation, correct and novel methodology, and thorough theoretical proof. However, the main weakness is that the experimental part is not sufficiently complete.

**Questions For Authors:**

The following questions are based on my previous comments, and I encourage the authors to carefully review them.

1. What are the details of the experiments, including the data generation process for the synthetic dataset and the implementation details of the methods?

2. In the related work section, the authors mention many related works, but why are many of them not compared in the experiments?

3. It would be better if suitable real-world datasets could be found to validate the effectiveness of the proposed method in real-world scenarios.

Overall, I appreciate the writing, motivation, methodology, and theory presented in this paper. If the authors could address the points mentioned above and make the experimental section more complete, I would be happy to increase my score.

**Relation To Broader Scientific Literature:**

The paper applies the Neyman regret theory from Multi-Armed Bandits (MAB) to adaptive ATE estimation. The relevant literature is correctly cited, and the authors clearly highlight the differences and advantages compared to these works. The discussion is also accurate and well-written.

**Theoretical Claims:**

The theoretical proof presented in the paper is detailed, clear, and well-reasoned.

---

> ### Author Rebuttal · Authors · 2025-04-01
>
> > It would be beneficial to compare the proposed method with some of the many related works, if possible.
>
> We did not compare our algorithm with Kato et al. (2020) because Cook et al. (2022) showed that ClipSDT significantly outperforms Kato’s algorithm, even under weaker assumptions (see Figure 2 in Cook et al.). Since we already compare against ClipSDT—the stronger of the two—it would be redundant to include Kato et al. (2020) in our evaluation.
>
> As for Kato et al. (2024), the setting considered in that work is different from ours: their focus is on adaptively selecting which covariates to observe, whereas our problem does not involve covariates at all. As such, their method is not applicable to the setting we study.
>
> > What are the details of the experiments, including the data generation process for the synthetic dataset and the implementation details of the methods?
>
> Each of our figures is based on simulations in a two-armed Bernoulli bandit setting, with different Neyman allocations. The mean rewards are set to $\mu_1 = 1/2$ and $\mu_0 \in \{0.05, 0.1, 0.2, 0.3, 0.4, 0.5\}$. At each round $t$, we sample $R_t(a) \sim \text{Bernoulli}(\mu_a)$ for $a \in \{0, 1\}$, and the algorithm observes the realized reward $R_t(A_t)$ corresponding to the chosen action $A_t$.
>
> This interaction continues for $T$ rounds, after which we compute the estimated ATE. We repeat this process across multiple independent runs and report the variance of the resulting ATE estimates.
>
> These experimental details are provided in lines 310–318 (right column) and lines 375–379.  We agree that some parameters were only implicitly specified in Figure 2, and have revised the experimental section to explicitly list all relevant parameters, including the exact values used for the Bernoulli means, the number of rounds, and the number of repetitions. We hope this improves the clarity and completeness of the experimental description.
>
> > the authors mention many related works, but why are many of them not compared in the experiments?
>
> We discuss this in the experiments section on Lines 303-307 (right column). Many of the prior works were designed for the IPW estimator, and so their performance in comparison to the AIPW estimator is so poor that it visually obscured the graphs. These existing algorithms do not converge to the asymptotically minimal variance.
>
>
> > It would also be helpful to include experiments on real-world datasets to validate the effectiveness of the method in real-world scenarios.
>
> We have run simulations using the macro-insurance dataset from Dai et al. and will include the figure in the camera ready if accepted. Please see the table presented in our response to Reviewer kjqu for a summary of these simulations.

---

> > ### Comment · Reviewer_Gq4L · 2025-04-02
> >
> > Thank you for the detailed response. My questions and concerns have been well addressed, and I will update my previous comments to indicate clear acceptance. If the paper is accepted, I suggest that the authors make the following revisions in the camera-ready version:
> >
> > 1. Briefly explain in the experimental section (e.g., after Lines 303-307) why ClipSDT was selected as the only baseline method—specifically because ClipSDT is a state-of-the-art method, and the authors chose to omit baselines that have been shown to perform worse than ClipSDT, as well as those that do not converge at all.
> >
> > 2. Revise the experimental section to explicitly list all relevant parameters, including the exact values used for the Bernoulli means, the number of rounds, and the number of repetitions.
> >
> > 3. Provide detailed information on the real-world dataset added during the rebuttal phase, including data sources, definitions of the treatment and outcome, sample size, as well as the experimental setup and results with analysis, in the experimental section.

---

### Official Review · Reviewer_PNDL · 2025-03-11

**Overall Recommendation:** 2

**Summary:**

The paper designs an adaptive algorithm, OPTrack, to estimate the average treatment effect (ATE) by integrating the augmented inverse probability weighting (AIPW) estimator with the optimism principle from multi-armed bandits. The paper claims that OPTrack adaptively balances the exploration of minimizing the variance of AIPW while simultaneously minimizing the Neyman regret. The paper shows that OPTrack achieves the state-of-the-art Neyman regret. Finally, the paper conducts a simulation study to demonstrate that OPTrack attains a lower MSE than other algorithms.

**Claims And Evidence:**

On line 144, what is the definition of  R(a)? It appears that the authors have only defined R_t(a). My best guess is that \( R(a) = E[R_t(a)] \), but this is not clearly stated. Since this quantity plays a critical role in defining the paper’s parameter of interest, the lack of clarity has made it difficult to follow the rest of the paper.

**Essential References Not Discussed:**

No.

**Experimental Designs Or Analyses:**

I have checked the simulation study in section 6.

**Methods And Evaluation Criteria:**

On line 135, the authors assume that the conditional expectations of the potential outcomes are fixed. I do not understand the implication of this assumption, as the conditional expectation is a function of past observations. Could the authors clarify what real-world scenarios this assumption is reasonable? The same concern applies to the conditional variance.

**Other Comments Or Suggestions:**

line 48: "The rest of our paper is organized as follows. The remainder of this paper is structured as follows."

line 160: "hestimator"

line 144-146: "The Neyman regret is simply the difference in the normalized MSE between the optimal variance and the MSE of the estimate produced by Alg." I believe the one being subtracted is the optimal variance.

**Other Strengths And Weaknesses:**

I believe this paper requires significant revision to clearly justify why the setting and assumptions are valid in real-world scenarios and to clarify why their theoretical results are exciting for a broader audience. Additionally, the paper should demonstrate that the proposed algorithm performs well on at least one real-world dataset.

**Questions For Authors:**

See above.

**Relation To Broader Scientific Literature:**

The paper's contribution is mainly theoretical. Although the authors claim that their algorithm is critical for applications like randomized clinical trials, regrettably, neither did they use this as a running example to explain the validity of their setting and assumptions, nor did they demonstrate their method on a real dataset. For example, for each round in a clinical trial, in addition to the treatment and reward, the doctor should have additional information like the patient's pre-treatment covariate and the outcome for other patients. Also, I'd love to understand how the assumptions claimed on line 135-136 make sense in a randomized control trial.

**Theoretical Claims:**

No, I did not check them carefully, as their correctness would not affect my decision.

---

> ### Author Rebuttal · Authors · 2025-04-01
>
> > On line 144, what is the definition of $R(a)$?
>
> We meant that $\Delta = \mathbb{E}[R_t(1) - R_t(0)]$ for any $t$ and we had omitted the subscript for brevity since the choice of $t$ is immaterial. However, we agree that this could be confusing. To clarify, we have revised the definition to $\Delta = r^\star(1) - r^\star(0)$, where $r^\star(a)$ was already defined just above line 135. This should eliminate the ambiguity.
>
> ### Discussion on conditionally fixed mean and variance assumption
>
> One concrete example where our assumption holds is the i.i.d. setting which is a standard assumption for RCTs [1]. More broadly, this assumption is either equivalent to or a generalization of those made in several prior works on adaptive experimental design (Kato et. al, Cook et. al, Neopane et. al) and is the standard assumption made in the bandit literature.
>
> We also emphasize that these assumptions are *necessary* in a meaningful sense: the ATE is defined as the difference in mean outcomes under treatment and control, and without our assuming these means are conditionally constant, the ATE is ill-defined. Similarly, the Neyman allocation depends on the variances of the potential outcomes, so if the variances change across rounds, we no longer have a meaningful way to define an optimal allocation. Therefore, this assumption represents the weakest possible condition under which adaptive ATE estimation remains well-defined.
>
>
> > The paper's contribution is mainly theoretical...
>
> We would like to clarify that our contributions are both theoretical and methodological. In addition to presenting new theoretical insights, we propose a novel algorithm that is not a minor variant of existing methods—it is based on distinct design principles. Our algorithm enjoys a greatly simplified analysis, achieves significantly stronger theoretical guarantees than prior work, and demonstrates superior empirical performance in simulations.
>
> While our long-term motivation is to inform the design of adaptive clinical trials, we acknowledge that our current setting does not include covariates. However, it does capture a realistic and meaningful intermediate scenario where outcomes from previously treated patients are observed and used to inform future treatment decisions. The challenge of performing nonasymptotic analysis in this setting—without covariates—is itself nontrivial and had not been fully addressed in prior work.
>
> Extending our framework to incorporate covariates is indeed an important next step toward practical applicability in real clinical settings. We view our current work as a foundational step toward that goal.
>
> > I believe this paper requires significant revision to clearly justify why the setting and assumptions are valid in real-world scenarios and to clarify why their theoretical results are exciting for a broader audience.
>
> We respectfully disagree with this statement, as we believe it reflects a misunderstanding of the paper’s scope and contributions. Our goal is not to present a fully deployable solution for real-world clinical trials, but rather to address a fundamental theoretical problem that remains unresolved—even in the simplest setting without covariates. It is worth emphasizing that randomized controlled trials can, and often do, proceed without incorporating covariates into the treatment assignment, especially in early-phase trials, low-data settings, or when covariate information is unavailable or unreliable. Our work aims to rigorously understand this foundational setting, which has practical relevance in such scenarios, before tackling the added complexity of contextual information.
>
> The setting and assumptions we consider are standard in the literature, have been adopted in prior foundational works. Our theoretical results are novel, conceptually clean, and advance the understanding of Neyman regret in nonasymptotic regimes. Importantly, these insights led to the development of a new algorithm that is not only theoretically sound but also empirically superior to existing methods.
>
> While we agree that real-world applicability in full generality—particularly in large-scale or personalized settings—requires the incorporation of covariates, we view our work as a necessary first step. We will revise the introduction to clarify the scope, highlight the novelty of our contributions, and emphasize the broader relevance of our results beyond any single application.
>
> > Additionally, the paper should demonstrate that the proposed algorithm performs well on at least one real-world dataset.
>
> We have run simulations using the macro-insurance dataset from Dai et al. and will include the figure in the camera ready if accepted. Please see the table presented in our response to Reviewer kjqu for a summary of these simulations.
>
> [1] Imbens, G. W., & Rubin, D. B. (2015). *Causal Inference for Statistics, Social, and Biomedical Sciences* (See chapters 1 and 6)

---

> > ### Comment · Reviewer_PNDL · 2025-04-02
> >
> > I thank the authors for addressing my questions and responding to my comments. I was not previously familiar with adaptive experimental design and have since spent some time reading the literature. I now have a better understanding of the paper's contribution. However, I am still curious about the practical relevance of this design. How commonly is this method used in real-world applications? Could the authors provide at least one real-world example where an experiment was actually designed in this way, specifically without using covariates? I remain somewhat skeptical about the value of improving the Neyman regret convergence rate if, in reality, such designs are rarely implemented.

---

> > > ### Author Response · Authors · 2025-04-02
> > >
> > > We thank the reviewer for their thoughtful engagement and for taking the time to explore the adaptive experimental design literature.
> > >
> > > > Could the authors provide at least one real-world example where an experiment was actually designed in this way, specifically without using covariates?
> > >
> > > To directly address your request, we provide a list (including several high-profile RCTs) where treatment assignment did not depend on covariates.
> > > RCTs without covariates are not uncommon, especially in early-phase clinical trials, public health interventions, and A/B testing environments—where covariate information may be unavailable, unreliable, or deliberately excluded to preserve simplicity or reduce bias.
> > >
> > > 1. West of Scotland Coronary Prevention Study [1]
> > > 2. ALLHAT trial [2]
> > > 3. WHI Dietary Modification Trial [3]
> > > 4. Systolic Blood Pressure Intervention Trial [4]
> > > 5. Multiple COVID-19 therapeutic trials [5] [6] [7]
> > > 6. Multivitamin supplementation for HIV-infected women [8]
> > > 7. Epilepsy surgery trial [9]
> > > 8. Nocturnal oxygen therapy for COPD [10]
> > >
> > > In addition, smaller tech companies often perform A/B tests (e.g., for user interfaces, ad formats, or recommendation systems) without incorporating covariates, as setting up the necessary infrastructure for covariate adjustment can be complex and resource-intensive. We have updated the manuscript to include these examples, which further highlights the utility of our work by illustrating concrete settings where our algorithm offers immediate practical relevance.
> > >
> > > > I remain somewhat skeptical about the value of improving the Neyman regret convergence rate if, in reality, such designs are rarely implemented.
> > >
> > > While our method is motivated by practical challenges in adaptive experimental design, we reemphasize that the primary goal is theoretical and methodological: to introduce a new algorithmic design principle (optimism for adaptive ATE estimation) and analyze it in the simplest noncontextual setting where precise, nonasymptotic analysis is feasible. **Indeed, this is the first finite-sample analysis of an adaptive ATE estimation procedure which is able to utilize reward estimation in any form.**
> > >
> > > Our main claim is not that our algorithm is immediately deployable for modern clinical trials (although as we mention above there are settings where our algorithm is immediately applicable). Rather, our contribution is foundational: we seek to clarify the theoretical landscape and provide clean tools and insights—like the role of optimism and variance-adaptive allocation—that can serve as building blocks for future work in more complex settings.
> > >
> > > To drive this point home, we offer an analogy to the multi-armed bandit (MAB) literature. There, foundational ideas such as Upper Confidence Bounds, Thompson Sampling, and Track-and-Stop were all initially developed and analyzed in the noncontextual finite-arm setting. These algorithms were later extended to contextual bandits and reinforcement learning by the broader community—often leveraging the insights and tools first developed in the simpler setting. While we do not claim that our algorithm will have the same level of impact, we view our contribution as a step in the same direction: introducing and analyzing a new principle that we hope will be generalized and built upon.
> > >
> > > We appreciate the reviewers curiosity, and hope this response clarifies both the practical grounding and the theoretical significance of our work.
> > >
> > > [1] Shepherd, J. et al. (1995). Prevention of coronary heart disease with pravastatin in men with hypercholesterolemia. New England Journal of Medicine
> > >
> > > [2] Officers, A. L. L. H. A. T. (2002). The Antihypertensive and Lipid-Lowering Treatment to Prevent Heart Attack Trial (ALLHAT).  Journal of the American Medical Association
> > >
> > > [3] Assaf, A. R. et al. (2016). Low-fat dietary pattern intervention and health-related quality of life: The Women’s Health Initiative randomized controlled dietary modification trial. Journal of the Academy of Nutrition and Dietetics
> > > [4] SPRINT Research Group. (2015). A randomized trial of intensive versus standard blood-pressure control. New England Journal of Medicine
> > >
> > > [5] Boulware, D. R. et al. (2020). A randomized trial of hydroxychloroquine as postexposure prophylaxis for Covid-19. New England journal of medicine
> > >
> > > [6] Beigel, J. H. et al. (2020). Remdesivir for the treatment of Covid-19. New England Journal of Medicine
> > >
> > > [7] Cao, B et al. (2020). A trial of lopinavir–ritonavir in adults hospitalized with severe Covid-19. New England journal of medicine
> > >
> > > [8] Fawzi, W. W. et al. (2004). A randomized trial of multivitamin supplements and HIV disease progression and mortality. New England Journal of Medicine
> > >
> > > [9] Wiebe, S. et al. (2001). A randomized, controlled trial of surgery for temporal-lobe epilepsy. New England Journal of Medicine
> > >
> > > [10] Chaouat, A. et al. (1999). A randomized trial of nocturnal oxygen therapy in chronic obstructive pulmonary disease patients. European Respiratory Journal

---

### Official Review · Reviewer_mBJ8 · 2025-03-15

**Overall Recommendation:** 4

**Summary:**

This study proposes optimistic algorithms for the adaptive estimation of the average treatment effect. While existing studies focus on clipping-based algorithms to stabilize performance, the present work develops a method by setting the treatment-assignment probability $\pi_t$ close to $1/2$. In addition, the authors provide theoretical guarantees regarding the Neyman regret.

**Claims And Evidence:**

The paper's central claim is that an optimistic strategy leads to sublinear (specifically, logarithmic) Neyman regret with respect to the optimal (minimum variance) allocation.
To support this claim, the authors provide:
- A precise definition of a Neyman regret benchmark comparing the proposed adaptive estimator to the optimal variance achievable by any estimator-allocation pair.
- A non-asymptotic analysis showing that, under mild regularity conditions, their method achieves $\widetilde{O}(\log T)$ regret instead of the linear-type regret often encountered by existing clipping-based approaches when measured against a stronger baseline.

**Essential References Not Discussed:**

None.

**Ethical Review Concerns:**

N/A.

**Experimental Designs Or Analyses:**

I appreciate the authors' efforts in providing experimental results. The experimental setup is designed effectively. Given that this paper is primarily theoretical and methodological in nature, the exact outcomes of the experiments are less critical to my overall evaluation. Still, the experiments serve as a useful demonstration of the proposed methods in practice.

**Methods And Evaluation Criteria:**

The proposed optimistic approach is well motivated. A similar idea has appeared in earlier studies, such as the mixed A2IPW in Kato et al. (2020), cited in this manuscript. However, those prior approaches were introduced primarily as heuristics and lacked theoretical guarantees, even though they performed well empirically. By contrast, the authors of this paper rigorously formulate the methodology and establish theoretical guarantees on the Neyman regret under their algorithm.

**Other Comments Or Suggestions:**

- I found the following concurrent work on arXiv. If you believe it is related to your paper and it is feasible, could you briefly discuss its contents in your draft?

Georgy Noarov, Riccardo Fogliato, Martin Bertran, and Aaron Roth, “Stronger Neyman Regret Guarantees for Adaptive Experimental Design.”

**Other Strengths And Weaknesses:**

N/A.

**Questions For Authors:**

N/A.

**Relation To Broader Scientific Literature:**

N/A.

**Theoretical Claims:**

The theoretical claims appear to be correct.

---

> ### Author Rebuttal · Authors · 2025-04-01
>
> Thank you for your positive and thoughtful review. We appreciate your feedback and are glad that you found our work worthy of acceptance.
>
> In response to your suggestion, we have incorporated a discussion of the concurrent work by Noarov et al. (“Stronger Neyman Regret Guarantees for Adaptive Experimental Design”) into our related works section. As this paper was posted on arXiv on 2/24/25—after the submission deadline—we were unable to include it in the original draft.
>
> At a high level, this paper studies the fixed-design/adversarial setting introduced by Dai et al., and shows that, under additional assumptions, the ClipOGD algorithm from that work can be refined to yield stronger regret guarantees. However, their approach still relies on an IPW-based estimator. As a result, when adopting the definition of Neyman regret used in our paper, their algorithm suffers linear Neyman regret due to the inherent bias of IPW estimators in adaptive settings.

---

### Official Review · Reviewer_kjqu · 2025-03-17

**Overall Recommendation:** 4

**Summary:**

This paper proposes an adaptive algorithm for estimating the Average Treatment Effect (ATE) in a two-arm setup. The authors introduce an Optimistic Policy Tracking (OPTrack) method that uses “optimism in the face of uncertainty” to allocate subjects so as to reduce the variance of their final ATE estimator. They compare it to prior clipping-based methods (e.g., ClipSMT, ClipSDT) and demonstrate both theoretical (logarithmic Neyman regret) and empirical improvements, especially in low- to moderate-sample settings.

## update after rebuttal

The authors have addressed my concerns, and I have updated my score from **3** (Weak Accept) to **4** (Accept) in light of the rebuttal. I remain enthusiastic about the potential and contribution of this work.

**Claims And Evidence:**

Overall, I find the core claims supported by evidence. The theoretical results that I checked (Lemma B.1 and Lemmas 5.1–5.3) appear sound. The experiments, while limited to purely simulated data, do illustrate the claimed finite-sample improvements.  The claims are substantiated, although clarity and exposition could be greatly improved (e.g., Section 3 should be more clear and define the variables accordingly, the definition of Neyman regret seems off in eq. (5), result of Lemma B.1 is central to the algorithm and should be moved to the main text, etc.).

**Essential References Not Discussed:**

From what I can see, the paper does cite the key references (Kato et al., Dai et al., Neopane et al.), which are the most relevant for adaptive two-armed ATE.

**Experimental Designs Or Analyses:**

The experiments use i.i.d. Bernoulli data with varying $\mu_0$​ and $\mu_1$​, evaluating ATE estimates via normalized MSE. While valid as a proof-of-concept, testing semi-synthetic data or broader distributions would be insightful. Still, the results support the authors' claim the OPTrack converges faster than prior methods in finite samples.

**Methods And Evaluation Criteria:**

The authors focus on synthetic Bernoulli experiments. This is understandable as a first demonstration, yet adding semi-synthetic data could make the empirical study more realistic (see Dai et al. as an example).

The evaluation metric is the Neyman regret, which is the difference between the actual variance of the final A2IPW estimator and the optimal minimal variance over all possible estimators and allocations, scaled by $T$. Conceptually, this is a relevant criterion for ATE estimation, although the paper’s definition of this regret is somewhat buried and needs clearer exposition in the main text (they do define it, but it may confuse readers unfamiliar with prior works like Dai et al. or Neopane et al.).

**Other Comments Or Suggestions:**

* Possibly move the essential bounding arguments (or at least a summary) from Appendix B into the main text.
* In Section 4.2, clarify how one obtains the “confidence sequence” for $\sigma$. Right now, we see a reference to Lemma B.1 but not enough main-text explanation.

**Other Strengths And Weaknesses:**

* Strengths:
	* The main contribution of an optimistic approach for adaptive ATE estimation is novel and interesting.
	* The finite-sample bounds showing $\mathcal{O}(\log T)$ Neyman regret vs. the best possible variance are strong.
	* Empirical evidence does match the theoretical claims, with clear gains over prior algorithms.
* Weaknesses:
	* Exposition is occasionally confusing. For instance, Section 3 and 4.2 might be clearer if the difference between Eq. (1) and Eq. (2) were fleshed out more (particularly explaining $\widehat{r}_t$​ in the A2IPW estimator).
	* The paper’s definition of Neyman regret in Section 3 (Eq. (5)) is somewhat rushed, with the first part of the definition including an implicit summation over time that the second part does not have.
	* Confidence sequences are introduced abruptly in Section 4.2, with the main details deferred to an appendix. A short, intuitive explanation of “what a confidence sequence is” might help.
	* The experiments are purely synthetic Bernoulli. It would be stronger if at least one semi-synthetic study were provided, referencing Dai et al.

Given these issues, I would label this a Weak Accept: the results are sound and interesting, but the writing needs work to be more accessible.

**Questions For Authors:**

* Is the population assumed identical at each round (i.i.d.)?
* How might one adapt OPTrack to the contextual setting such as in Kato et al.?

**Relation To Broader Scientific Literature:**

The work is situated well among the literature on adaptive ATE estimation (Kato et al., Dai et al., Neopane et al.) and improves on prior designs that yield only asymptotic efficiency or use IPW-based regret, while the proposed OPTrack obtains non-asymptotic efficiency vs. the best A2IPW baseline. However, the lack of contextual or covariate-based experiments is a gap that might be addressed in future work (as Kato et al. do incorporate covariates).

**Theoretical Claims:**

* I skimmed the proofs of Lemma B.1 and Lemmas 5.1–5.3: they appear correct, relying on time-uniform concentration inequalities (as in “confidence sequences”) plus bounding $\|\sigma-\widehat{\sigma}\|$.
* I did not check every detail of the larger bounding arguments in the supplementary. However, nothing I saw raised red flags about correctness.
* One place for improvement would be bringing some of the main arguments from Appendix B into the main text—particularly the statement about how confidence sequences for standard deviations lead to bounding the Neyman allocation.

---

> ### Author Rebuttal · Authors · 2025-04-01
>
> > The claims are substantiated, although clarity and exposition could be greatly improved (e.g., Section 3 should be more clear and define the variables accordingly, the definition of Neyman regret seems off in eq. (5), result of Lemma B.1 is central to the algorithm and should be moved to the main text, etc.).
>
> We agree Section 3 needed clearer definitions and better exposition. We’ve revised it to improve clarity and precision.
>
> Regarding eq. (5), we understand the concern about the scaling mismatch. However, this scaling is intentional: multiplying the algorithm’s MSE term, $(\widehat \Delta_T - \Delta)^2$, by $T$ enables a direct comparison with the oracle’s MSE. We’ve clarified this by revising eq. (5), defining Neyman regret explicitly as the normalized excess MSE over the optimal policy, and recasting eq. (6) as a proposition to emphasize its connection.
>
> We agree that more discussion on Lemma B.1 is warranted and have moved it to Section 5 and expanded the discussion to highlight how it supports Lemmas 5.1 and 5.2.
>
> > The authors focus on synthetic Bernoulli experiments. This is understandable as a first demonstration, yet adding semi-synthetic data could make the empirical study more realistic (see Dai et al. as an example).
>
> We have run additional experiments using the macro-insurance intervention dataset used by Dai et al and will include them in the camera ready version. Our experiments on this real world dataset qualitatively align with our synthetic experiments. Below is a table summarizing the performance of various algorithms on this dataset.
>
> | Algorithm                  | 100    | 300    | 500    | 1000   | 1500   |
> | -------------------------- | ------ | ------ | ------ | ------ | ------ |
> | ClipSDT                    | 0.1178 | 0.1021 | 0.0987 | 0.0939 | 0.0956 |
> | OPTrack                    | 0.1036 | 0.0954 | 0.0942 | 0.0939 | 0.0923 |
> | Est. Reward Oracle                    | 0.1046 | 0.0974 | 0.0947 | 0.0934 | 0.0942 |
> | Oracle | 0.0924 | 0.0906 | 0.0932 | 0.0908 | 0.0910 |
>
> > However, the lack of contextual or covariate-based experiments is a gap that might be addressed in future work (as Kato et al. do incorporate covariates).
>
> We agree this is an important direction. Kato et al. and Cook et al. address contextual settings but rely on asymptotic analysis. In contrast, our work is the first to provide nonasymptotic guarantees for adaptive reward estimation, even in the simpler noncontextual case. We see this as a foundational step toward handling contextual settings in a rigorous, finite-sample framework.
>
> > The main contribution of an optimistic approach for adaptive ATE estimation is novel and interesting.
>
> We appreciate the recognition of our optimistic approach. In addition, our work is, to our knowledge, the first to address adaptive reward estimation with finite-sample guarantees—even without covariates. Our method also yields strong empirical performance while enjoying a simpler and more transparent analysis.
>
> > Exposition is occasionally confusing. For instance, Section 3 and 4.2 might be clearer if the difference between Eq. (1) and Eq. (2) were fleshed out more (particularly explaining $\widehat r_t$ in the A2IPW estimator).
>
> Thank you for flagging this. We previously explained some differences between Eq. (1) and (2) in lines 154–161 but have now expanded this explanation for clarity. The distinctions are as follows:
> - Eq. (1) uses marginal action probabilities $\mathbb{P}(A_t)$, while Eq. (2) uses conditional probabilities $\pi_t(A_t)$ based on past data.
> - The standard AIPW estimator uses data-splitting, while A2IPW requires only that $\widehat r_t$ be predictable (i.e., adapted to the past). Beyond this, $\widehat r_t$ may depend freely on past data. We have added this explanation after Eq. (2).
> We also corrected a typo in Eq. (2) where $(r)$ should have been $(\hat{r})$.
>
> > Is the population assumed identical at each round (i.i.d.)?
>
> Our assumption is more general than i.i.d. As noted in lines 133–136, we assume only that the means and variances of the potential outcomes are conditionally fixed over time. Other aspects of the data generating process may vary arbitrarily across rounds. This form of conditional stationarity is necessary for the ATE and Neyman allocation (and hence the problem) to be well defined.
>
> > How might one adapt OPTrack to the contextual setting such as in Kato et al.?
>
> We agree this is a compelling direction for future exploration. Our primary goal here was to establish a rigorous nonasymptotic analysis of adaptive reward estimation in the simplest (tabular) setting, as no prior work had done so. Extending OPTrack to incorporate covariates, as in Kato et al., would introduce new complexities in controlling reward estimation errors across various function classes, providing a meaningful and challenging path for future research.

---

> > ### Comment · Reviewer_kjqu · 2025-04-04
> >
> > Dear authors,
> >
> > Thank you for your thoughtful and thorough rebuttal. You have addressed all of my concerns, and I have updated my score accordingly. I continue to be very enthusiastic about this work.

---

### Decision · Program_Chairs · 2025-05-01

**Decision:**

Accept (poster)

**Comment:**

The reviewers generally acknowledged the strengths of the work and are satisfied with the authors’ rebuttals. I concur with the reviewers’ opinions and recommend acceptance of this work. However, the reviewers also raised several points for improvement, particularly concerning the numerical experiments and the clarity of exposition. The authors should incorporate the reviewers’ comments and further improve their paper in the camera-ready version.